# Evaluation of internal coupling and coordination degree and diagnosis of obstacle factors for high-quality regional economic development: Evidence from Chongqing's " One District, Two Groups"

**Sifang Che**[1]*, **Xi Zhang**[1], **Weijia Shu**[2]

1 Chongqing-Chengdu Shuangcheng Economic Circle Construction Research Institute, Chongqing Technology and Business University, Chongqing, China, 2 School of Economics, Chongqing Technology and Business University, Chongqing, China

* 903755722@qq.com

**Data Availability Statement:** All relevant data are within the manuscript and its Supporting Information files.

## Abstract

The study builds a high-quality development index system encompassing dimensions of economic vitality, coordinated development, green development, and digital development. Utilizing the entropy weight TOPSIS method, the coupling coordination model, and the barrier factor model, this research examines the high-quality economic development level, internal coupling coordination, and obstacle factors in Chongqing's "one district and two groups" from 2017 to 2021. The findings are as follows: (1)The high-quality economic development level of the "one district and two groups" has been increasing year by year. This growth is driven by the joint progress across all dimensions, though regional disparities exist in the development levels of different dimensions. The order of high-quality economic development from highest to lowest is: the main city metropolitan area, southeast Chongqing, and northeast Chongqing. (2)The development level of internal coupling coordination for high-quality economic development within the "one district and two groups" has also been rising annually. However, the overall level of coupling coordination development remains low, although regional coupling coordinated development is gradually forming. There was no evidence of σ convergence in the coupling coordination of high-quality economic development within the "one district and two groups." The main city metropolitan area shows stability with an expansion trend, while the "two groups" exhibit divergence. (3)Overall, Chongqing's barrier index order was "economic vitality > coordinated development > green development > digital development." There were differences in obstacle factors between "one district" and "two groups"; the "two groups" align with the overall performance, while the main metropolitan area performs as "green development > economic vitality > coordinated development > digital development." The research conclusions provide a theoretical basis and decision-making reference for improving coordinated development within the region. (4)The main obstacles to high-quality economic development in Chongqing and the

**Funding:** This work was supported by the Key Project of the National Social Science Youth Foundation project (Grant No.: 21CTJ007), the Chongqing Natural Science Foundation Project (Grant No.: cstc2021jcyj-bshX0123), the Humanities and Social Sciences Fund Project of Chongqing Municipal Education Commission (Grant No.: 21SKGH114), and the Key Research Platform Open Project for Chongqing Technology and Business University (Grant No.: KFJJ2019030).There was no additional external funding received for this study.

**Competing interests:** The authors have declared that no competing interests exist.

"one district and two groups" are centered on the dimensions of economic vitality and coordinated development, while also reflecting green development considerations.

## 1. Introduction

High-quality economic development is a significant marker of China's economic progress in the new era. With the rapid advancements in globalization, the knowledge economy, and the network economy, high-quality regional economic development plays a crucial role in narrowing regional development gaps and achieving common prosperity. However, the imbalance and inadequacy in development across regions severely hinder high-quality economic growth. Therefore, promoting high-quality regional economic development has become an urgent issue that needs addressing. Currently, the academic community has conducted numerous studies on the connotation, level measurement, and driving factors of high-quality regional economic development. High-quality economic development is a multi-dimensional and complex system. Achieving this requires enhancing the development vitality of economic entities, fostering coordinated development among various regional factors, and advancing the green and digital transformation of the development model. Research indicates that subsystems are interrelated rather than isolated, and it is essential to promote the coordinated development of these coupled subsystems. Most studies focus on the coupling development between different factor systems, such as the relationship between regional economic growth and the environment, ecological protection and the economy, technological innovation and economic quality, and the digital economy and high-quality development [1–3]. However, few studies explore the internal coupling and coordinated development of subsystems within high-quality economic development, and there is a lack of research on diagnosing the obstacle factors. As can be seen, few studies have explored the coupled and coordinated development among the subsystems within the high-quality development of the economy, and there is a lack of research on the diagnostic aspects of its obstacle factors. Meanwhile, Chongqing Municipality, as one of the important regional economic centers in the western region, its special strategic position of location advantage makes the central government has always attached great importance to the economic and social development of Chongqing, as an important strategic pivot for the development of western China, which bears the great responsibility of promoting the development of the western region. As a matter of fact, since its direct administration in 1997, Chongqing Municipality has made great progress in all aspects of economy, society and culture. By the end of 2023, Chongqing Municipality's GDP amounted to 3014.5 billion yuan, an increase of nearly 20 times compared with 1997, surpassing Guangzhou and becoming the 4th city in China's urban GDP ranking. Currently, Chongqing, as the core area of the Chengdu-Chongqing Twin Cities Economic Circle, plays the role of an engine in promoting the formation of China's economic growth "fourth pole". However, Chongqing is a collection of "big cities, big rural areas, big mountainous areas, big reservoirs" in one, the inter-regional factor endowment differences are large, resulting in the unbalanced and inadequate development of its various regions is particularly prominent. In order to realize the integrated and coordinated development of the region, Chongqing Municipality has put forward the strategic layout of "one district and two groups", i.e., to take the main city metropolitan area as the core, and to drive the coordinated development of the town clusters of the Three Gorges Reservoir Area in the northeast part of Chongqing, and the town clusters of the Wuling Mountain Area in the southeast part of Chongqing, the implementation of which has a typical demonstration effect in

promoting the coupling and coordinated development of the internal region. However, the "one region and two clusters" of Chongqing Municipality face a series of obstacles in the process of development, which have a constraining effect on the coupling and coordination of the regional economic system, thus affecting the process of high-quality development of the regional economy. In view of this, this paper takes "one region and two clusters" of Chongqing Municipality as the research object, constructs a multidimensional evaluation index system that affects the high-quality development of the economy, adopts the entropy weight method to measure its development level, and introduces the coupling coordination degree model and obstacle factor model to focus on the development level of the coupling coordination degree of the high-quality development of the regional economy, as well as the obstacle factors to measure and analyze the development level of the coupling coordination degree of the high-quality development of the regional economy. Measured and analyzed, it has important theoretical and practical value for promoting the high-quality development of regional economy in Chongqing Municipality and even China as a whole.

Under the comparison of existing studies, the marginal contribution of this paper is mainly reflected in the following three aspects: (1) Starting from the internal system of high-quality regional economic development, it constructs a multi-dimensional evaluation system covering economic vitality, digital development, green development, coordinated development, etc., which provides new perspectives and methods for assessing high-quality regional economic development. (2) The coupling and coordinated development mechanism between the internal subsystems of high-quality regional economic development is explored in depth, providing a solid theoretical foundation for promoting the high-quality of coordinated development within the regional economy. (3) The convergence and obstacles of coupling and coordination in high-quality regional economic development are precisely analyzed, and the direction of improving high-quality regional economic development is clarified, providing a scientific basis for policy formulation.

## 2. Literature review

### 2.1 Connotation analysis of high-quality economic development

High-quality economic development was rich in connotation. From the perspective of quality, efficiency, fairness, and sustainable development, it involves high resource allocation efficiency, low costs for resources and the environment, and positive economic and social benefits [4]. It emphasizes quality and efficiency to achieve higher quality, more efficient, fairer, and more sustainable development [5, 6]. Based on the five new development concepts, high-quality development is rooted in innovation, green, coordination, openness, and sharing [7]. Hong Yinxing points out that these concepts are core to high-quality development: innovation is the driving force, coordination is the form, green is the inherent requirement, openness is the mechanism for internal and external linkage, and sharing is the fundamental purpose [8]. Other perspectives include Ren Baoping, who views high-quality development as involving effectiveness, coordination, innovation, sustainability, and sharing [9]. Zhao Jianbo et al. suggest that high-quality development must be comprehensive and balanced, prioritizing people's livelihood [10].

### 2.2 Evaluation of high-quality economic development

This research focuses on measuring development levels, analyzing regional differences, and evaluating coupling coordination. High-quality economic development is assessed at national [11, 12], provincial [13, 14], and regional levels [15, 16]. The evaluation involves constructing index systems, selecting weights, and measuring high-quality economic development indices.

Currently, the index system is mainly constructed based on researchers' understanding of the connotation of high-quality economic development, and some scholars have constructed a high-quality economic development evaluation index system in accordance with the "five new development concepts" and sustainable development [17–19], which meets the requirements of economic development in the new era, but needs to be improved to reflect the actual high-quality economic development, which fits the requirements of economic development in the new era, but there is still room for improvement in reflecting the actual high-quality development of the economy; and some researchers measure high-quality development of the economy from the level of efficiency, for example, some studies have used labor productivity [20], total factor productivity [21–23] or GDP per capita [24] to measure the high quality of the economy, and these indicators can effectively characterize the level of high-quality economic development from a certain perspective, but they are still not comprehensive enough. With the depth of research, scholars have found that innovation-driven [25–27], green finance [28, 29], digital economy [27, 30], environmental regulation [31, 32] and other factors can promote high-quality economic development, while resource dependence can seriously hinder high-quality economic development [25]. Therefore, these factors should also be taken into consideration when designating the evaluation index system for high-quality economic development. Various methods are used to select index weights: equal weight method (easy to calculate but lacks differentiation), subjective methods (expert scoring, hierarchical analysis, influenced by human factors), objective methods (entropy weight method, artificial neural networks, principal component analysis, reducing human influence but sometimes misaligned with real-world importance), and combined methods (AHP-EVM, expert evaluation-BP method, combining subjective and objective advantages but requiring consistency tests). Regional difference analysis and coupling coordination degree evaluations reveal disparities and their sources using Dagum Gini coefficient [16, 33], Thiel Index [34, 35], and vertical-horizontal methods [36]. Coupling coordination measures the interaction impact between systems, such as scientific and technological innovation with high-quality economic development [37, 38] or the digital economy with high-quality economic development [3].

### 2.3 Evaluation of high-quality economic development

While significant progress has been made in understanding the connotation and evaluation of high-quality economic development, several issues remain. Studies have focused on national, provincial, or regional levels, with little attention to intra-provincial regional differences. There is no consensus on the construction of evaluation index systems or index weight selection methods. The academic community has examined the coupling coordinated development between economic quality development systems and other systems but lacks focus on the internal coupling coordination within economic quality development systems. Internal coordinated development is vital for addressing regional imbalances. Researchers have extensively studied the factors driving high-quality economic development [39–41] but have rarely analyzed the obstacle factors hindering the coordinated development of internal subsystems. Therefore, this study focuses on measuring high-quality economic development levels, evaluating internal coupling coordination, and diagnosing obstacle factors to provide a theoretical basis for enhancing high-quality regional economic development.

## 3. Data, indicators, and methods

### 3.1 Data source and description

This paper mainly measures and analyzes the internal coupling coordination level and obstacle factors of the high-quality economic development of "one district and two groups" in

Chongqing. Chongqing's "one district and two groups" was chosen as the object of study because of the uniqueness and importance of this strategy for the harmonious development of Chongqing's regional economy. As an important city in the development of western China, Chongqing's "one district and two groups" strategy aims to optimize the regional economic structure, promote regional integration, and achieve efficient allocation of resources and balanced economic development. The "one district" refers to the main urban area of Chongqing, which is the political, economic and cultural center of Chongqing and has a strong radiation and driving effect on the development of the surrounding areas. By improving the core competitiveness of the main city, it can effectively promote the regional economic agglomeration and diffusion effect. The "two groups" refer to the town clusters in the Three Gorges Reservoir Area in northeast Chongqing and the town clusters in the Wuling Mountain Area in southeast Chongqing, which are tasked with promoting local economic development while protecting the ecological environment. Through the development of these two regions, the industrial division of labor and complementary advantages within the Chongqing region can be realized, and the balanced development of the regional economy can be promoted. The implementation of this strategy will help to play the role of radiation driven by the main city of Chongqing, promote the economic development of the surrounding areas, but also help to alleviate the pressure of the development of the main city, to achieve balanced regional development. In addition, Chongqing's "one district and two groups" strategy is also closely related to the construction of the Chengdu-Chongqing Twin-city Economic Circle. The Chengdu-Chongqing Twin Cities Economic Circle is an important economic growth pole in western China, and its development is of great significance in promoting the economic development of western China. Through the strategy of "one district and two groups", Chongqing can strengthen cooperation with neighboring cities such as Chengdu, jointly promote the harmonious development of the regional economy and form new economic growth points. Chongqing's "one region, two groups" strategy also focuses on green and innovative development, which will help promote high-quality economic development. By optimizing the industrial structure and developing high-tech industries and modern service industries, Chongqing can achieve sustainable economic development while protecting the ecological environment. In summary, Chongqing's "one region, two groups" strategy is not only an important measure to promote the harmonious development of the regional economy, but also an effective way to achieve high-quality economic development. The implementation of this strategy will help elevate Chongqing's position in the overall development of the country and inject new vitality into the economic development of western China.

Therefore, the research data mainly come from Chongqing Statistical Yearbook, Chongqing Statistical Yearbook of Chongqing districts and counties, Chongqing Statistical Bulletin of National Economic and Social Development, Chongqing Statistical Bulletin of National Economic and Social Development of Chongqing districts and counties, EPS Global Statistical Data Platform, China Economic Big Data Research Platform, etc. Partial missing data were completed by the interpolation method.

### 3.2 Data source and description

**3.2.1 Selection of indicators.** High-quality economic development is the substantive stage of China's current economic development. Since 2017, China's economy has shifted from a stage of high-speed growth to a stage of high-quality economic development. At present, the academic community has not reached a unified opinion on the connotation of high-quality economic development, nor has it formed a unified standard for the evaluation index system of high-quality economic development. Based on the principles of combining

**Table 1. Evaluation index system for high-quality development of regional economy.**

| Dimension | Metric | Indicator interpretation (unit) | Attribute |
|---|---|---|---|
| **Economic vigor** | Per capita GDP | GDP / Total population (yuan / person) | + |
| | All-personnel labour productivity | GDP / year (RMB / person year) | + |
| | The added value of industry | The newly added value in the production process of enterprises with an annual main income of more than 20 million yuan (10,000 yuan) | + |
| | The added value of the financial industry accounted for it | Financial sector value added / GDP (%) | - |
| | Foreign trade dependence degree | Total import and exports / GDP (%) | + |
| | Asset-liability ratio | Total liabilities / total assets of (%) | - |
| | R&D funding investment intensity | R&D expenditure / GDP | + |
| | Patent authorization | The sum of the three patents granted for invention, practical novelty and design (piece) | + |
| | Density of population | Total population / area (person / km$^2$) | + |
| | The number of teachers with compulsory education is available per 10,000 people | Number of full-time teachers in compulsory education / 10,000 | + |
| | Jobless rate | Urban registered unemployment rate (%) | - |
| **Harmonious development** | The proportion of tertiary industry | Output value of the tertiary industry / GDP (%) | + |
| | Advanced industry | Output value of tertiary industry / output value of secondary industry | + |
| | Product sales rate | Industrial sales output value / total industrial output value: 100% | + |
| | Per capita disposable income ratio of urban and rural residents | Per capita disposable income of urban residents / per capita disposable income of rural residents | - |
| | The Engel coefficient ratio of urban and rural residents | The Engel coefficient of urban residents / that of rural residents | + |
| | Urbanization rate | Urban population / total population | + |
| | The proportion of education expenditure | Education expenditure / general public finance budget expenditure | + |
| | The proportion of financial deposit balance | Financial institution deposit balance / GDP | + |
| | The proportion of financial loan balance | Financial institution loan balance / GDP | + |
| **Green development** | Air quality | Air quality rate | + |
| | | Atmospheric fine particles PM10 (microgram / m$^3$) | - |
| | | Sulfur dioxide emissions (micrograms / m$^3$) | - |
| | Land area covered with trees | Forest area / land area of (%) | + |
| | Yearly precipitation | Total annual precipitation (mm) | + |
| | Groundwater resources amount | The sum of various groundwater of useful value (one hundred million cubic meters) | + |
| | Regional environmental noise | Provisions on the allowable range of noise in order to protect the health and living environment of the population (decibel) | - |
| | Energy consumption | Total industrial energy consumption (10,000 tons of standard coal) | - |
| **Digital development** | Level of digitization | Level of digitization | + |
| | Digital Inclusive Finance | Digital Inclusive Finance Development Index | + |
| | TV development | TV coverage is (%) | + |

innovation and history, typicality and availability, independence and reality, this paper constructs the index system of "one district and two groups" in Chongqing, as shown in Table 1.

**3.2.2 Indicator instructions.** (1) Economic vigor. economic vigor is an important indicator of the rate of growth of aggregate supply and demand in the economy of a country or region over a given period of time, as well as its potential. It involves not only macroeconomic indicators such as the growth rate of gross national product, the fixed investment rate and the savings rate, but also the capacity of urban economic development, such as the ability to bring in capital and attract high-quality labor. Economic vitality is closely related to high-quality economic development, because high-quality development is not only quantitative growth, but

more importantly qualitative improvement, including the optimization of economic structure [42], the transformation and upgrading of industries [43], and the coordinated development of urban and rural areas [44], among other aspects. In the process of promoting high-quality economic development, the measurement of economic vitality is particularly important. It can help us identify and solve the structural contradictions and institutional mechanism obstacles that constrain economic development, and promote economic restructuring and the transformation of old and new kinetic energy.

(2) Harmonious development. harmonious development is an important criterion for measuring high-quality economic development, this is because it embodies the comprehensiveness and balance of economic development, and ensures the coordinated progress of various fields, regions and social strata. It emphasizes improving social well-being and the ecological environment while enhancing economic indicators, promoting balanced development among different regions, industries and social groups, ensuring rational use of resources and intergenerational equity, enhancing the economy's innovative and risk-resistant capabilities, and achieving economic and social development goals through policy guidance [45–47]. Coordinated development helps improve the country's international competitiveness and promotes long-term stable economic growth and comprehensive social progress. In the context of the new era, China's economy has shifted from the stage of high-speed growth to the stage of high-quality development, a shift that requires the economy not only to pursue quantitative growth, but also to focus on quality improvement and structural optimization.

(3) Green development. Green development is closely related to high-quality economic development, which is not only an important content of high-quality development, but also the key to promoting economic transformation and upgrading. The core of green development is to achieve economic growth, improve resource utilization efficiency, reduce pollution emissions, and promote the virtuous cycle of socio-economic and natural ecosystems under the premise of protecting the environment [48, 49]. At the policy level, the Chinese government has taken green development as a national strategy to promote a comprehensive green transformation of economic and social development by formulating relevant policies and standards. For example, it promotes the green and low-carbon transformation and upgrading of traditional industries, vigorously develops green and low-carbon industries, accelerates the synergistic transformation and development of digitalization and greening, and steadily promotes the green and low-carbon transformation of energy. These measures are aimed at building a green, low-carbon and recycling economic system, realizing a steady decline in carbon emissions after reaching the peak, and laying a solid foundation for building a beautiful China. In addition, green development involves the optimization of spatial patterns, such as optimizing the pattern of territorial spatial development and protection, creating green development highlands, and building a spatial pattern of green, low-carbon and high-quality development. These measures help to form green modes of production and lifestyles, improve resource utilization efficiency, and reduce environmental pollution, thus promoting high-quality economic development.

(4) Digital development. A large number of studies have shown that the digital economy can effectively promote the high-quality development of the economy [50, 51]. Digital development can significantly improve production efficiency, promote innovation, enhance market responsiveness, and promote industrial upgrading and regional coordinated development [52]. Digital transformation injects new momentum into economic growth by optimizing resource allocation, modernizing the industrial chain, improving the efficiency of government services, strengthening data security governance, promoting green development, enhancing international competitiveness, and facilitating the optimization of employment structure [53].

It also helps to build a modernized economic system, enhance the country's global competitiveness and support the overall progress of society.

## 3.3 Research methods

**3.3.1 Measurement method of high-quality economic development level.** The selection of appropriate methods is the basic premise of accurately measuring the high-quality economic development level. At present, the academic circle has carried out many studies and discussions on the measurement method of high-quality economic development. In fact, measuring the level of high-quality development of the economy is mainly divided into three steps: the first step is to build an evaluation index system, see Table 1. The second step is to select the weight of each index. Through comparison, this paper selects the entropy right TOPSIS method to select the weight of each index. The method will be introduced in detail below. The third step is to add up, mainly the use of index data and weight plus assembly economy high quality development index or score.

Entropy right TOPSIS method combines entropy weight method with TOPSIS method, which is a mathematical method of multi-objective optimization and has strong advantages in index evaluation. In fact, the entropy weight method determines the weight of the index according to the influence of the relative degree of change on the whole, which can avoid the interference of human factors. At the same time, the TOPSIS method is a commonly used group comprehensive evaluation method, can make full use of the original data information, the results can accurately reflect the gap between the evaluation scheme, it is mainly used to study the evaluation object and the distance of the "ideal solution", combined with "ideal solution" (positive and negative ideal solution), calculate the final close *C* value, to quantitative sorting. The core of entropy right TOPSIS method is TOPSIS, but in the calculation, the entropy value (entropy weight method) should be used to calculate the weight of each evaluation index, and the evaluation index data and the weight to get new data, and then the new data is used to study the TOPSIS method. Therefore, this paper uses the entropy right TOPSIS method to measure the high quality development level of Chongqing economy. The specific steps are described as follows:

**Table 2. Evaluation index system for high-quality development of regional economy.**

| Metric | Weight | Metric | Weight |
|---|---|---|---|
| Per capita GDP | 0.0263 | Urbanization rate | 0.0246 |
| All-personnel labour productivity | 0.0191 | The proportion of education expenditure | 0.0118 |
| Value added | 0.1309 | The proportion of financial deposit balance | 0.0452 |
| The added value of the financial industry accounted for it | 0.0295 | The proportion of financial loan balance | 0.0805 |
| Foreign trade dependence degree | 0.1075 | Air quality rate | 0.0055 |
| Asset-liability ratio | 0.0161 | Atmospheric inhalable particulate matter PM10 | 0.0060 |
| R&D funding investment intensity | 0.0407 | Sulfur dioxide emissions | 0.0027 |
| Patent authorization | 0.0687 | Land area covered with trees | 0.0122 |
| density of population | 0.1588 | Yearly precipitation | 0.0158 |
| The number of teachers with compulsory education is available per 10,000 people | 0.0097 | Groundwater resources amount | 0.0481 |
| Urban registered unemployed number | 0.0028 | Regional environmental noise | 0.0083 |
| The proportion of tertiary industry | 0.0148 | Energy consumption | 0.0028 |
| Advanced industry | 0.0551 | Level of digitization | 0.0097 |
| Product sales rate | 0.0030 | Digital Inclusive Finance Development Index | 0.0200 |
| Per capita disposable income ratio of urban and rural residents | 0.0143 | TV coverage is (%) | 0.0029 |
| The Engel coefficient ratio of urban and rural residents | 0.0063 | | |

Standardization. The raw data was standardized using the extreme standardization method. Assuming that there are n evaluation objects and m evaluation indicators, the following raw data matrix $X$ is constructed:

$$X = \begin{Bmatrix} x_{11} & x_{12} & \dots & x_{1m} \\ x_{21} & x_{22} & \dots & x_{2m} \\ \dots & \dots & \dots & \dots \\ x_{n1} & x_{n2} & \dots & x_{nm} \end{Bmatrix}_{n \times m} \tag{1}$$

$X$ Standardizing on the original data matrix, a new matrix can be obtained. $R = (r_{ij})_{n \times m}$, $r_{ij} \in [0,1]$, To represent the standard value of the first evaluation object on the first evaluation index.

$$\text{Forward indicators} : r_{ij} = \frac{x_{ij} - \min x_{ij}}{\max x_{ij} - \min x_{ij}} \tag{2}$$

$$\text{Negative indicators} : r_{ij} = \frac{\max x_{ij} - x_{ij}}{\max x_{ij} - \min x_{ij}} \tag{3}$$

Construct a weighted matrix $Z$. Suppose $P_{ij}$ is the proportion of the $j$ evaluation index of the $i$ evaluation object of the matrix $R$, have $P_{ij} = \frac{r_{ij}}{\sum\limits_{i=1}^{n} r_{ij}}$. Suppose $e_j = -\frac{1}{\ln n} \sum\limits_{i=1}^{n} P_{ij} \ln P_{ij}$ represent the entropy value of the index $j$, then the weight of each evaluation index, from which the weighted matrix can be obtained: $w_j = \frac{(1 - e_j)}{\sum\limits_{j=1}^{m} (1 - e_j)}$

$$Z = w_j \times R = \begin{Bmatrix} w_1 r_{11} & w_2 r_{12} & \dots & w_m r_{1m} \\ w_1 r_{21} & w_2 r_{22} & \dots & w_m r_{2m} \\ \dots & \dots & \dots & \dots \\ w_1 r_{n1} & w_2 r_{n2} & \dots & w_m r_{nm} \end{Bmatrix} = \begin{Bmatrix} z_{11} & z_{12} & \dots & z_{1m} \\ z_{21} & z_{22} & \dots & z_{2m} \\ \dots & \dots & \dots & \dots \\ z_{n1} & z_{n2} & \dots & z_{nm} \end{Bmatrix} \tag{4}$$

The optimal scheme $Z^+$ and the worst scheme $Z^-$ are determined by the weighted matrix. Order $z_j^+$ and $z_j^-$ represent the maximum and minimum values of the indicator $j$ in all evaluated objects. There are:

$$\text{optimal solution} : Z^+ = \begin{pmatrix} z_1^+ & z_2^+ & \dots & z_m^+ \end{pmatrix} \tag{5}$$

$$\text{the worst solution} : Z^- = \begin{pmatrix} z_1^- & z_2^- & \dots & z_m^- \end{pmatrix} \tag{6}$$

Calculate the Euclidean distance between each evaluation object $Z^+$ and the sum $Z^-$.

$$D_i^+ = \sqrt{\sum_{j=1}^{m} (z_{ij} - z_j^+)^2} \tag{7}$$

$$D_i^- = \sqrt{\sum_{j=1}^{m} (z_{ij} - z_j^-)^2} \tag{8}$$

Calculate the proximity between each evaluation object and the optimal scheme.

$$C_i = \frac{D_i^-}{D_i^- + D_i^+} \tag{9}$$

The larger the value $C_i$, the higher the high quality level of economic development of the evaluation object $i$, the worse the versa. Therefore, the size of the value $C_i$ is the ranking of the high quality economic development level of each district and county in Chongqing.

After calculation, the weight of each index is shown in Table 2.

**3.3.2 Evaluation method of coupling and coordination degree.** The degree of coordination is used to measure the degree of harmony between the system or the internal elements of the system in the development process, reflecting the trend of the system from disorder to order, and is a quantitative indicator of the degree of coordination status [54]. The degree of coupling coordination can reflect the level of coordinated development and virtuous cycle between the system or elements, and can directly judge the interaction type between the system by its numerical value, so as to facilitate the comprehensive evaluation and study of the whole system. In fact, the economic quality development index system of economic vitality, coordinated development, green development and digital development is not isolated four systems, there are multiple coupling relationship between subsystems, using coupling coordination model to explore the coupling coordination relationship between them, to reveal the internal connection between subsystems, the exploration of Chongqing economic development of high quality. Therefore, referring to the practice of Wang Shujia et al. [55], the coupled coordination model of high-quality economic development in Chongqing was constructed, and the coupling coordination degree between the high-quality economic development subsystems of "one district and two Groups" in Chongqing was measured, so as to provide reference for further improving the benign and coordinated development among the subsystems of high-quality economic development. In this paper, the coupled coordination model of four dimensions of "one district and two groups" in Chongqing is constructed as follows:

$$C = 4 \times \frac{\sqrt[4]{Q_1 \times Q_2 \times Q_3 \times Q_4}}{Q_1 + Q_2 + Q_3 + Q_4} \tag{10}$$

$$D = \sqrt{C \times T} \tag{11}$$

$$T = \alpha Q_1 + \beta Q_2 + \gamma Q_3 + \delta Q_4 \tag{12}$$

Among them, $Q_1$ to $Q_4$ represents the economic vitality level, coordinated development level, green development level and digital development level respectively; C is the coupling degree, and its value is between 0–1. The closer the C value is to 1, the better the coupling state between systems; the closer the C value is to 0, the worse the coupling state between systems. D is the coupling and coordination degree, and T is the comprehensive evaluation index of the four dimensions of high-quality economic development. $\alpha,\beta,\gamma,\delta$ were the undetermined coefficient, because the weight of each dimension is calculated according to the entropy method, it is set as $\alpha = 0.6102, \beta = 0.2557, \gamma = 0.1014\ \delta = 0.0327$. Referring to the existing research literature, this paper takes 0.1,0.2,0.3,0.4,0.5,0.6,0.7,0.8,0.9,1.0 as the separation points, dividing the coupling coordination level into extreme disorder, severe disorder, moderate disorder, mild disorder, near disorder, barely coordination, primary coordination, intermediate coordination, good coordination, good coordination, and high coordination.

**3.3.3 Diagnosis method of disorder factors.** To explore the obstacles of Chongqing "one district and two groups" high quality economic development main obstacle factor, further

clear "area two groups" of Chongqing economic development of high quality internal subsystem coordinated development relations improvement direction, this paper refer to Kuang Lihua et al. [56], the introduction of barriers model, diagnosis of the Chongqing "one district and two groups" economic high quality internal coupling coordination obstacle factor, specific calculation formula is as follows:

$$O_{ij} = \frac{(1 - X_{ij}) \times w_{ij}}{\sum (1 - X_{ij}) \times w_{ij}} \tag{13}$$

$$U_i = \sum O_{ij} \tag{14}$$

In particular, $O_{ij}$ represents the obstacle degree of the j-th second level indicator in the first level indicator i to the internal coupling and coordination relationship of high-quality economic development, then $U_i$ represents the obstacle degree of the first level indicator i; $X_{ij}$ represents the standardized value of the j-th secondary indicator, and $1-X_{ij}$ represents the degree of deviation of the indicator; $w_{ij}$ is the weight of the j-th indicator.

## 4. Results

### 4.1 High-quality economic development level and its characteristic distribution of "one district and two groups"

In this study, the high-quality economic development level of "one district and two groups" in Chongqing from 2017 to 2021 was accurately measured, and the results are shown in Table 3.

According to Table 3, the high-quality economic development level of "one district and two groups" in Chongqing showed an increasing trend from 2017 to 2021. Among them, the high-quality development level index of the main urban area is the highest, and the main reason for the high-quality economic development level of the main urban area is that the development level of the central city is very high, and the development level of the new urban area is slightly higher than that of southeast Chongqing and northeast Chongqing. At the same time, the high-quality economic development level of the Wuling mountain area in southeast Chongqing is slightly higher than that of the Three Gorges Reservoir area in northeast Chongqing.in this regard, The level of high-quality economic development in the central urban area increased from 0.2454 in 2017 to 0.2839 in 2021, The average annual growth rate was 3.14%; The main city new area increased from 0.0976 in 2017 to 0.1470 in 2021, The average annual growth rate reached 10.12%; Growth from 0.1609 in 2017 to 0.2056 in 2021, The average annual growth rate reached 5.56%; While the urban group of Wuling Mountain area in southeast Chongqing increased from 0.0964 in 2017 to 0.1259 in 2021, The average annual growth rate was 6.12%; The town group of the Three Gorges Reservoir area in northeast Chongqing increased from 0.0947 in 2017 to 0.1226 in 2021, The average annual growth rate was 5.89%. It can be seen that from the perspective of growth rate, the main city new area is faster than

**Table 3. High-quality economic development of "One District and Two Groups" in Chongqing from 2017 to 2021.**

| Area | 2017 | 2018 | 2019 | 2020 | 2021 |
|---|---|---|---|---|---|
| Main city metropolitan area | 0.1609 | 0.1769 | 0.1863 | 0.1910 | 0.2056 |
| Central urban area | 0.2454 | 0.2679 | 0.2616 | 0.2630 | 0.2839 |
| Main city new area | 0.0976 | 0.1087 | 0.1298 | 0.1370 | 0.1470 |
| Northeast Chongqing three Gorges Reservoir area town group | 0.0947 | 0.0929 | 0.1023 | 0.1137 | 0.1226 |
| Southeast Chongqing Wuling mountain area town group | 0.0964 | 0.1104 | 0.1069 | 0.1273 | 0.1259 |

southeast Chongqing, southeast Chongqing is faster than northeast Chongqing, northeast Chongqing is faster than the main city metropolitan area, and the main city metropolitan area is faster than the central city.

At the same time, in order to investigate the influencing factors of the high-quality economic development of "one district and two groups", this study also measured the development of each dimension, and the results are shown in Table 4.

Obviously, it can be found from Table 4 that the economic vitality, coordinated development, green development and digital development levels of the main city metropolitan area, southeast Chongqing and northeast Chongqing are all increasing from 2017 to 2021. Obviously, the main urban metropolitan area has a high level in three dimensions: economic vitality, coordinated development and digital development; the urban group in northeast Chongqing, but lags behind the main urban area, which is the reason why the high quality development level of the main urban metropolitan area is much higher than the urban group in northeast Chongqing and Wuling Mountain area in southeast Chongqing. Specifically, one is the main urban area economic vitality development level is far higher than the northeast and southeast Chongqing, according to the calculation, urban urban area economic vitality development level is about 2.8 times of "two groups"(Calculated from the data in Table 3), which means that "two groups" and urban urban area of economic development vitality gap is still larger, narrow the "two groups" and "area" economic vitality gap become the important path of Chongqing regional economic development of high quality. Second, the coordinated development level of downtown metropolitan area is still higher than the northeast and southeast, and downtown metropolitan area coordinated development level is about 1.7 times of "two groups", although the gap is slightly less than the gap of economic vitality, but promote the coordinated development level of "two groups" is one of the key factors to promote the development of Chongqing economy high quality. Three is the urban metropolitan area green development level is significantly lower than the northeast and southeast of Chongqing, "two groups" of green development level is about more than 2 times of urban metropolitan area, visible, urban metropolitan area green development level is restricting the development of economic quality, how to improve the urban metropolitan area green development level become the current and future priority. Four is although the main metropolitan area of digital

**Table 4. Table of the development of each dimension of "One District and two Groups" in Chongqing municipality[a].**

| Dimension | Region | 2017 | 2018 | 2019 | 2020 | 2021 |
|---|---|---|---|---|---|---|
| Economic vigor | Main city metropolitan area | 0.0907 | 0.1025 | 0.1056 | 0.1065 | 0.1199 |
| | Northeast Chongqing | 0.0321 | 0.0373 | 0.0368 | 0.0395 | 0.0422 |
| | Southeast Chongqing | 0.0341 | 0.0385 | 0.0375 | 0.0407 | 0.0454 |
| Harmonious development | Main city metropolitan area | 0.0581 | 0.0608 | 0.0608 | 0.0640 | 0.0661 |
| | Northeast Chongqing | 0.0323 | 0.0363 | 0.0393 | 0.0425 | 0.0444 |
| | Southeast Chongqing | 0.0333 | 0.0355 | 0.0413 | 0.0447 | 0.0492 |
| Green development | Main city metropolitan area | 0.0210 | 0.0226 | 0.0258 | 0.0303 | 0.0293 |
| | Northeast Chongqing | 0.0447 | 0.0373 | 0.0410 | 0.0500 | 0.0543 |
| | Southeast Chongqing | 0.0450 | 0.0525 | 0.0464 | 0.0605 | 0.0536 |
| Digital development | Main city metropolitan area | 0.0122 | 0.0159 | 0.0225 | 0.0219 | 0.0241 |
| | Northeast Chongqing | 0.0065 | 0.0095 | 0.0171 | 0.0164 | 0.0178 |
| | Southeast Chongqing | 0.0064 | 0.0105 | 0.0175 | 0.0173 | 0.0186 |

[a]In the table, northeast Chongqing refers to the three Gorges Reservoir area town group in northeast Chongqing, and southeast Chongqing refers to the town group of Wuling Mountain area in southeast Chongqing, the same below.

**Table 5. Score of high-quality economic development level of "one district and two groups" in Chongqing from 2017 to 2021.**

| Area | 2017 | 2018 | 2019 | 2020 | 2021 |
|---|---|---|---|---|---|
| Main city metropolitan area | 80.1701 | 87.2539 | 91.4056 | 93.5101 | 96.5700 |
| Northeast Chongqing town group | 50.7763 | 50.0000 | 54.1737 | 59.2318 | 63.1816 |
| Southeast Chongqing town group | 51.5195 | 57.7489 | 56.2163 | 65.2248 | 64.6098 |

development level is higher than the northeast and southeast of Chongqing, but relative to other dimensions, "area" and "two groups" the smallest digital development gap, this may be the Chongqing digital development overall level is not too high, accelerate the digital development is still a reliable path to promote the development of Chongqing economy of high quality.

In addition, in order to reflect the spatial distribution characteristics of the high quality development level of "one district and two groups" in Chongqing from 2017 to 2021, the score of the high quality development level of "one district and two groups" was also measured. The results are shown in Table 5.

As can be seen from Table 5, during the five years from 2017 to 2021, "one district and two groups" steadily promoted the construction of high-quality economic development, and the respective high-quality level steadily increased. Specifically, the high-quality development level of the main city metropolitan area is higher than that of the northeast Chongqing town group and the southeast Chongqing town group, which has reached a high level of development in 2019. The development of the town group in northeast Chongqing is almost the same as that in southeast Chongqing, but the town group in southeast Chongqing has achieved a medium level of development in 2020, which is faster than the town group in northeast Chongqing.

## 4.2 Results and analysis of the coupling and coordination degree level

Based on the coupling coordination method and classification standard, the level of high-quality economic development of "one district and two groups" in Chongqing was measured, and the results are shown in Table 6.

As can be seen from Table 6, from 2017 to 2021, the development level of the internal coupling coordination degree of high-quality economic development of "one district and two Groups" in Chongqing is increasing year by year. The level of the coupling coordination degree is much higher than that of southeast Chongqing and northeast Chongqing, and southeast Chongqing is slightly higher than that of northeast Chongqing. Specifically, the coupling coordination value of the main metropolitan area increased from 0.501 in 2017 to 0.854 in

**Table 6. Coupled and coordination level and grade of high-quality economic development of "One District and Two Groups" in Chongqing from 2017 to 2021.**

| Year | Region | Main city metropolitan area | Northeast Chongqing | Southeast Chongqing |
|---|---|---|---|---|
| 2017 | Coupling coordination degree value | 0.501 | 0.173 | 0.223 |
| | Coupling coordination level | Forced coordination | Major maladjustment | Moderate dysregulation |
| 2018 | Coupling coordination degree value | 0.648 | 0.303 | 0.437 |
| | Coupling coordination level | Primary coordination | Mild dysregulation | On the verge of dysregulation |
| 2019 | Coupling coordination degree value | 0.763 | 0.454 | 0.501 |
| | Coupling coordination level | Intermediate coordination | On the verge of dysregulation | Forced coordination |
| 2020 | Coupling coordination degree value | 0.822 | 0.543 | 0.612 |
| | Coupling coordination level | Good coordination | Forced coordination | Primary coordination |
| 2021 | Coupling coordination degree value | 0.854 | 0.603 | 0.649 |
| | Coupling coordination level | Good coordination | Primary coordination | Primary coordination |

2021, with an average annual growth rate of 14.09%; the coupling coordination degree of urban groups in northeast Chongqing increased from 0.173 in 2017 to 0.603 in 2021, with an average annual growth rate of 49.71%; and the coupling coordination degree of urban groups in Wuling Mountain area in southeast Chongqing increased from 0.223 in 2017 to 0.649 in 2021, with an average annual growth rate of 38.21%. It can be seen that from the perspective of the coupling and coordination growth rate of high-quality economic development of "one district and two groups", northeast Chongqing is higher than southeast Chongqing, and "two groups" is significantly higher than the main city metropolitan area, which also shows that the gap between the internal coupling and coordination degree of high-quality economic development of "one district and two groups" in Chongqing is gradually narrowing. In addition, from the point of regional coupling coordination level, although "one district and two groups" economic development of high quality internal coupling coordination from disorder to coordination, but coordination level is low, in addition to the main city to achieve good coordination in 2021, northeast and southeast Chongqing in the primary coordination, it also shows that the overall coordination is still low, accelerate the economic high quality coordinated development is still an important task of the current and future.

## 4.3 Convergence analysis of coupling and coordinated development

From Table 6, we can find that there are some differences in the coupling and coordination degree of high-quality economic development between "one district and two groups" in Chongqing. In order to further explore the characteristics of these differences, this paper draws on the practice of Hua Jian et al. [37], and uses σ convergence analysis on the coupling coordination degree of "one district and two groups". The calculation formula is:

$$\sigma_t = \left\{ N^{-1} \sum_{m=1}^{N} \left[ X_m(t) - [N^{-1} \sum_{k=1}^{N} X_k(t)] \right]^2 \right\}^{\frac{1}{2}} \tag{15}$$

Among, $X_m(T)$ indicates the coupling coordination degree of the m district and counties in year t; N, represents the number of districts and counties, N = 38 (i. e. the number of districts and counties in Chongqing).like $\sigma_{t+1}$ less-than $\sigma_t$ It shows that there is σ convergence, which indicates that the gap between the coupling and coordination degree of high-quality economic development in various districts and counties shows a narrowing trend. According to the above formula and related conditions, the measurement results are shown in Fig 1.

As can be intuitively seen from Fig 1, from the perspective of the whole city, the σ value of Chongqing is stable from 2017 to 2020, and increased slightly in 2021, which means that the coupling and coordination of high-quality economic development of Chongqing as a whole may show an expanding trend in the future. By region, the σ value trend of the σ value of the urban area in northeast Chongqing showed an inverted U-shaped development trend, showing a decreasing trend from 2017 to 2018–2021. The σ value of the urban group in Wuling mountain area in southeast Chongqing showed an alternating pattern of "rise-down-up", increasing from 2017 to 2018, decreasing from 2018–2019, and increasing from 2019–2021. On the whole, the coordination of high-quality economic development of "one district and two groups" shows a divergent trend, indicating that the coordination of high-quality economic development of districts and counties within "one district and two groups" does not show a decreasing trend, but shows a trend of expansion. At the same time, the overall σ value of the whole city is slightly higher than that of the main urban area, but the σ value of the first two is significantly higher than that of southeast Chongqing and northeast Chongqing.

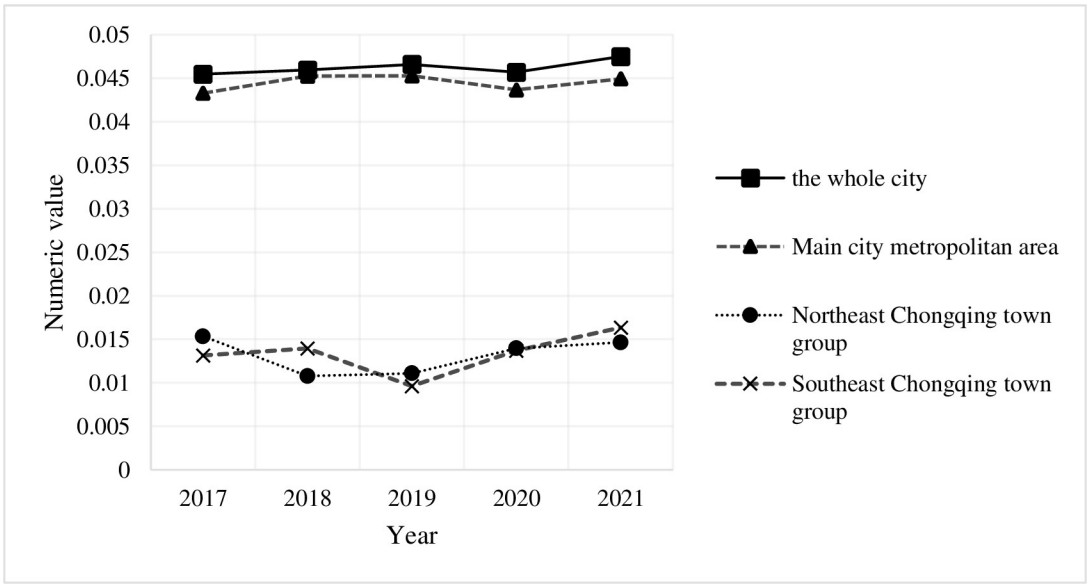

**Fig 1. Convergence test diagram of the coupling and coordination degree of high-quality economic development of "One District, Two Groups" in Chongqing.**

## 4.4 Diagnosis and analysis of obstacle degree factor results

On the basis of the overall evaluation mentioned above, we diagnose the obstacle factors affecting the high-quality development of Chongqing's economy, and identify the main obstacle factors according to the criterion-level indicators and indicator-level indicators, and the results are shown in Tables 7–10.

**4.4.1 Guideline level analysis.** It can be found from Table 7 that in the five years of 2017 to 2021, the level of obstacles index is ranked as "economic vitality > coordinated development > green development > digital development". The higher the degree of obstacles means, the heavier the impact of this index, indicating that economic vitality plays a very important role in the high-quality economic development of Chongqing.

From Table 8 can be found, 2017–2021, southeast Chongqing and northeast Chongqing economic development of high quality level index barriers the results consistent with the overall situation in Chongqing, namely "economic development > green development > digital development", shows that the key factors hindering the development of "two groups" is still economic vitality, coordinated development, green development of the third. However, in the main urban area, there is "green development > economic vitality > coordinated development > digital development", which means that the green development index is the key factor hindering the development of the main urban area, and the economic vitality is the

**Table 7. Results of high-quality development barriers at the guideline level in Chongqing from 2017–2021.**

| Subject investigated | Economic vigor | Harmonious development | Green development | Digital development |
|---|---|---|---|---|
| In 2017, the barrier degree mean value | 0.5394 | 0.2344 | 0.2037 | 0.0224 |
| In 2018, the barrier degree mean value | 0.5366 | 0.2312 | 0.1947 | 0.0374 |
| In 2019, the barrier degree mean value | 0.5484 | 0.2693 | 0.1562 | 0.0261 |
| In 2020, the barrier degree mean value | 0.5493 | 0.2613 | 0.1657 | 0.0237 |
| In 2021, the barrier degree mean value | 0.5479 | 0.2664 | 0.1601 | 0.0256 |

**Table 8. Results of high-quality development barriers at the guideline level in "One District and two Groups" from 2017–2021.**

| Year | Area | Economic vigor | Harmonious development | Green development | Digital development |
|---|---|---|---|---|---|
| 2017 | Main city metropolitan area | 0.2294 | 0.2071 | 0.5635 | 0 |
| 2017 | Northeast Chongqing | 0.6964 | 0.2534 | 0.0229 | 0.0273 |
| 2017 | Southeast Chongqing | 0.6925 | 0.2428 | 0.0247 | 0.0400 |
| 2018 | Main city metropolitan area | 0.2416 | 0.1993 | 0.5101 | 0.0490 |
| 2018 | Northeast Chongqing | 0.6635 | 0.2407 | 0.0587 | 0.0370 |
| 2018 | Southeast Chongqing | 0.7048 | 0.2536 | 0.0154 | 0.0263 |
| 2019 | Main city metropolitan area | 0.2024 | 0.3922 | 0.4055 | 0 |
| 2019 | Northeast Chongqing | 0.6814 | 0.2428 | 0.0390 | 0.0369 |
| 2019 | Southeast Chongqing | 0.7615 | 0.1729 | 0.0242 | 0.0414 |
| 2020 | Main city metropolitan area | 0.2173 | 0.3560 | 0.4267 | 0 |
| 2020 | Northeast Chongqing | 0.6718 | 0.2438 | 0.0461 | 0.0383 |
| 2020 | Southeast Chongqing | 0.7586 | 0.1843 | 0.0244 | 0.0327 |
| 2021 | Main city metropolitan area | 0.2103 | 0.3701 | 0.4196 | 0 |
| 2021 | Northeast Chongqing | 0.6837 | 0.2547 | 0.0229 | 0.0387 |
| 2021 | Southeast Chongqing | 0.7498 | 0.1745 | 0.0377 | 0.0380 |

second, and the coordinated development index is the third. At the same time, it is worth noting that the digital development barriers of the main metropolitan area were zero in the five years from 2017 to 2021, which indicates that the digital development of the main metropolitan area is at a high level and will not restrict the high-quality economic development of the main metropolitan area.

**4.4.2 Indicator level analysis.** Due to the large number of indicators in the indicator layer, according to the size of the obstacle degree, this paper only lists the top seven factors in the ranking of obstacle degree, and the results are shown in Table 9.

**Table 9. Results of high-quality development barriers at the indicator level in Chongqing from 2017–2021.**

| Year | Event | Ranking of barrier factors | | | | | | |
|---|---|---|---|---|---|---|---|---|
| | | 1 | 2 | 3 | 4 | 5 | 6 | 7 |
| 2017 | Barrier factors | Density of population | The added value of industry | Groundwater resources amount | Foreign trade dependence degree | The proportion of financial loan balance | Patent authorization | Advanced industry |
| | Barriers | 0.1307 | 0.1029 | 0.1025 | 0.0804 | 0.0539 | 0.0538 | 0.0451 |
| 2018 | Barrier factors | Density of population | The added value of industry | Groundwater resources amount | Foreign trade dependence degree | The proportion of financial loan balance | Patent authorization | Advanced industry |
| | Barriers | 0.1285 | 0.1012 | 0.1004 | 0.0801 | 0.0548 | 0.0538 | 0.0451 |
| 2019 | Barrier factors | Density of population | The added value of industry | Advanced industry | Groundwater resources amount | Foreign trade dependence degree | Patent authorization | The proportion of financial loan balance |
| | Barriers | 0.1340 | 0.1104 | 0.0826 | 0.0825 | 0.0811 | 0.0558 | 0.0539 |
| 2020 | Barrier factors | Density of population | The added value of industry | Groundwater resources amount | Foreign trade dependence degree | Advanced industry | The proportion of financial loan balance | Patent authorization |
| | Barriers | 0.1350 | 0.1109 | 0.0867 | 0.0772 | 0.0725 | 0.0574 | 0.0546 |
| 2021 | Barrier factors | Density of population | The added value of industry | Groundwater resources amount | Advanced industry | Foreign trade dependence degree | Patent authorization | The proportion of financial loan balance |
| | Barriers | 0.1358 | 0.1117 | 0.0815 | 0.0779 | 0.0759 | 0.0557 | 0.0533 |

**Table 10. Results of high-quality development barriers at the indicator level in "One District and two Groups" from 2017–2021.**

| Year | Region | Event | Ranking of barrier factors | | | | | | |
|---|---|---|---|---|---|---|---|---|---|
| | | | 1 | 2 | 3 | 4 | 5 | 6 | 7 |
| 2017 | Main city metropolitan area | Barrier factors | Groundwater resources amount | Yearly precipitation | Asset-liability ratio | Per capita disposable income ratio of urban and rural residents | Land area covered with trees | All-personnel labour productivity | The proportion of education expenditure |
| | | Barriers | 0.3015 | 0.0990 | 0.0926 | 0.0897 | 0.0765 | 0.0760 | 0.0740 |
| 2017 | Northeast Chongqing | Barrier factors | Density of population | The added value of industry | Foreign trade dependence degree | The proportion of financial loan balance | Patent authorization | Advanced industry | R&D funding investment intensity |
| | | Barriers | 0.1860 | 0.1388 | 0.1321 | 0.0990 | 0.0723 | 0.0638 | 0.0493 |
| 2017 | Southeast Chongqing | Barrier factors | Density of population | The added value of industry | Foreign trade dependence degree | Patent authorization | Advanced industry | The proportion of financial loan balance | The proportion of financial deposit balance |
| | | Barriers | 0.2060 | 0.1698 | 0.1089 | 0.0891 | 0.0715 | 0.0627 | 0.0587 |
| 2018 | Main city metropolitan area | Barrier factors | Groundwater resources amount | All-personnel labour productivity | Yearly precipitation | Per capita disposable income ratio of urban and rural residents | Asset-liability ratio | Land area covered with trees | The proportion of education expenditure |
| | | Barriers | 0.2707 | 0.1075 | 0.0889 | 0.0805 | 0.0795 | 0.0687 | 0.0664 |
| 2018 | Northeast Chongqing | Barrier factors | Density of population | The added value of industry | Foreign trade dependence degree | The proportion of financial loan balance | Patent authorization | Advanced industry | R&D funding investment intensity |
| | | Barriers | 0.1777 | 0.1322 | 0.1263 | 0.0946 | 0.0715 | 0.0631 | 0.0478 |
| 2018 | Southeast Chongqing | Barrier factors | Density of population | The added value of industry | Foreign trade dependence degree | Patent authorization | Advanced industry | The proportion of financial loan balance | The proportion of financial deposit balance |
| | | Barriers | 0.2078 | 0.1713 | 0.1139 | 0.0899 | 0.0721 | 0.0697 | 0.0592 |
| 2019 | Main city metropolitan area | Barrier factors | Groundwater resources amount | Advanced industry | All-personnel labour productivity | Per capita disposable income ratio of urban and rural residents | Asset-liability ratio | Yearly precipitation | Land area covered with trees |
| | | Barriers | 0.2397 | 0.1831 | 0.0841 | 0.0713 | 0.0699 | 0.0631 | 0.0608 |
| 2019 | Northeast Chongqing | Barrier factors | Density of population | The added value of industry | Foreign trade dependence degree | The proportion of financial loan balance | Patent authorization | Advanced industry | R&D funding investment intensity |
| | | Barriers | 0.1774 | 0.1462 | 0.1262 | 0.0945 | 0.0703 | 0.0647 | 0.0478 |
| 2019 | Southeast Chongqing | Barrier factors | Density of population | The added value of industry | Foreign trade dependence degree | Patent authorization | The proportion of financial loan balance | The proportion of financial deposit balance | R&D funding investment intensity |
| | | Barriers | 0.2244 | 0.1850 | 0.1170 | 0.0971 | 0.0672 | 0.0639 | 0.0566 |
| 2020 | Main city metropolitan area | Barrier factors | Groundwater resources amount | Advanced industry | All-personnel labour productivity | Yearly precipitation | Per capita disposable income ratio of urban and rural residents | Asset-liability ratio | Land area covered with trees |
| | | Barriers | 0.2435 | 0.1530 | 0.0967 | 0.0800 | 0.0724 | 0.0715 | 0.0618 |
| 2020 | Northeast Chongqing | Barrier factors | Density of population | The added value of industry | Foreign trade dependence degree | The proportion of financial loan balance | Patent authorization | Advanced industry | R&D funding investment intensity |
| | | Barriers | 0.1783 | 0.1459 | 0.1261 | 0.0944 | 0.0658 | 0.0646 | 0.0462 |
| 2020 | Southeast Chongqing | Barrier factors | Density of population | The added value of industry | Foreign trade dependence degree | Patent authorization | The proportion of financial loan balance | The proportion of financial deposit balance | R&D funding investment intensity |
| | | Barriers | 0.2266 | 0.1868 | 0.1055 | 0.0980 | 0.0778 | 0.0645 | 0.0581 |

(*Continued*)

**Table 10.** (Continued)

| Year | Region | Event | Ranking of barrier factors | | | | | | |
| --- | --- | --- | --- | --- | --- | --- | --- | --- | --- |
| | | | 1 | 2 | 3 | 4 | 5 | 6 | 7 |
| 2021 | Main city metropolitan area | Barrier factors | Groundwater resources amount | Advanced industry | All-personnel labour productivity | Yearly precipitation | Asset-liability ratio | Per capita disposable income ratio of urban and rural residents | Land area covered with trees |
| | | Barriers | 0.2394 | 0.1683 | 0.0906 | 0.0786 | 0.0714 | 0.0712 | 0.0607 |
| 2021 | Northeast Chongqing | Barrier factors | Density of population | The added value of industry | Foreign trade dependence degree | The proportion of financial loan balance | Patent authorization | Advanced industry | R&D funding investment intensity |
| | | Barriers | 0.1801 | 0.1478 | 0.1273 | 0.0954 | 0.0687 | 0.0653 | 0.0462 |
| 2021 | Southeast Chongqing | Barrier factors | Density of population | The added value of industry | Foreign trade dependence degree | Patent authorization | The proportion of financial deposit balance | The proportion of financial loan balance | R&D funding investment intensity |
| | | Barriers | 0.2272 | 0.1873 | 0.1003 | 0.0983 | 0.0647 | 0.0646 | 0.0582 |

From Table 9, it can be found that during the period of 2017–2021, although there are changes in the ranking results of the barrier degree of the indicator layer in Chongqing, it is always the seven barrier factors of density of population, the added value of industry, groundwater resources amount, foreign trade dependence degree, the proportion of financial loan balance, patent authorization and advanced industry, which are mainly reflective of the development of the three dimensions of economic vigor, harmonious development, and green development. Specifically, as a populous city, the population density of Chongqing directly affects urban planning, infrastructure construction and public service provision, which in turn affects the stable and efficient operation of the economy; industry is an important pillar of Chongqing's economy, and the growth of the value added of industry directly reflects the efficiency of industrial production and the degree of optimization of industrial structure, which is fundamental to the high-quality development of the economy; water resources are one of the key factors constraining the economic and social development of the region; and water resources are one of the key factors constraining the economic and social development of the region. Water resources are one of the key factors constraining the economic and social development of the region. The amount of underground water resources in Chongqing is directly related to the safety of urban water supply, industrial and agricultural water demand and ecological environmental protection; Chongqing as an inland open highland, the dependence on foreign trade reflects the degree of openness and international competitiveness of its economy, and has an important impact on the diversification and stability of the economy; finance is the core of the modern economy, and the ratio of the balance of financial loans reflects the strength of financial support to the real economy, which is crucial for promoting industrial upgrading and economic restructuring. industrial upgrading and economic restructuring; the number of patents granted is an important indicator of innovation capacity, reflecting the level of scientific and technological innovation and industrial competitiveness of a region, which plays an important role in promoting high-quality development of the economy; industrial advancement represents the optimization and upgrading of the industrial structure, and it is an important symbol of high-quality development of the economy, which involves the modernization, high-endization and intelligence of the industrial chain.

From Table 10, it can be found that during the period of 2017–2021, the top seven obstacles to the high-quality development of the economy of the northeast Chongqing include density of population, the added value of industry, foreign trade dependence degree, the proportion of financial loan balance, patent authorization, advanced industry and R&D funding investment

intensity. During 2017–2018, the top seven obstacle factors affecting the high-quality development of the economy of the southeast Chongqing include density of population, the added value of industry, foreign trade dependence degree, patent authorization, advanced industry, the proportion of financial loan balance, and the proportion of financial deposit balance, while after 2018, advanced industry exits the top seven and is replaced by R&D funding investment intensity. Although there are differences in specific obstacle factors, the results of the obstacle degree in the indicator layer of high-quality economic development in southeast Chongqing and northeast Chongqing are mainly reflected in the dimensions of economic vigor and harmonious development, indicating that the key factors for the development of the "two groups" are still economic vigor and harmonious development. It is worth noting that during the period of 2017–2021, the total proportion of population density and industrial value added obstacles in the obstacle factors of the two groups exceeds 30%, indicating that population density and industrial value added are the main constraints to the high-quality economic development of the two groups. This is because the lower population density in the "two groups" region leads to limited market size, insufficient consumer demand and labor supply, thus restricting the high-quality development of the economy. At the same time, the weak industrial base and low value-added of industry in the northeastern Chongqing and southeastern Chongqing make it difficult to drive the rapid growth of the regional economy and optimize and upgrade the industrial structure, thus affecting the high-quality development of the economy. During the period 2017–2021, the top seven barriers to high-quality economic development in the main city metropolitan area are still mainly reflecting the three dimensions of economic vigor, harmonious development and green development, although there are changes in the top seven barriers.

## 5. Conclusions and revelation

This paper measures and analyzes the internal coupling coordination level and obstacle factors of the high-quality economic development in Chongqing's "one district and two groups." The research builds a regional economic development evaluation system with 31 indicators, covering dimensions such as economic vitality, coordinated development, green development, and digital development. Using the entropy weight TOPSIS method, the study assesses the economic development level of Chongqing's "one district and two groups" in the new era and applies the coupling coordination model and barrier factor model to analyze the internal coupling coordination development and identify the obstacle factors restricting high-quality economic development. The research findings indicate: (1)From 2017 to 2021, the high-quality economic development level of Chongqing's "one district and two groups" increased yearly, driven by the joint growth of all dimensions. The main urban area's development index was the highest, with the Wuling Mountain area in southeast Chongqing slightly surpassing the Three Gorges Reservoir area in northeast Chongqing. The growth rates showed that the main city new area developed fastest, followed by southeast Chongqing, northeast Chongqing, and finally the central city. While economic vitality, coordinated development, green development, and digital development levels increased, regional differences remained in various dimensions. (2)The internal coupling and coordination degree of high-quality economic development in Chongqing's "one district and two groups" also improved yearly. The metropolitan area's level far exceeded that of southeast and northeast Chongqing, with southeast Chongqing slightly ahead of northeast Chongqing. From 2017 to 2021, the main city metropolitan area progressed from forced coordination to good coordination, northeast Chongqing from serious imbalance to primary coordination, and southeast Chongqing from moderate imbalance to primary coordination, indicating a gradual formation of internal coupling and coordinated development. However, convergence analysis showed that the coupling coordination degree of high-

quality economic development in the "one district and two groups" diverged, indicating a widening trend rather than convergence. (3)Overall, the obstacle index in Chongqing is "economic vitality > coordinated development > green development > digital development," underscoring the critical role of economic vitality in high-quality economic development. Differences exist between the main urban area and the "two groups," with the "two groups" following the overall pattern while the main urban area prioritizes "green development > economic vitality > coordinated development > digital development." (4)According to the results of the analysis of the indicator level, Chongqing and the main city metropolitan area should focus on the development of the three dimensions of economic vigor, coordinated development, and green development, while the northeastern and southeastern parts of Chongqing should focus on economic vigor and coordinated development.This suggests that Chongqing's high-quality economic development should pay attention to harmonious coexistence with the environment while cultivating economic vitality and promoting coordinated development.

Based on these findings, the paper offers several policy recommendations to enhance high-quality regional economic development: (1)Continuously Promote High-Quality Regional Economic Development: While Chongqing's "one district and two groups" have improved yearly, this trend should be further consolidated. By fostering development in all sectors, a strong synergy can push high-quality economic development to new heights. (2)Leverage the Leading Role of the Main City Metropolitan Area: As the forefront of Chongqing's high-quality economic development, the main city metropolitan area should continue enhancing its development level, providing reference experiences and models for the "two groups." The "two groups" should also leverage their advantages to accelerate development and catch up. (3) Implement Differentiated Development Strategies: Given the varying development levels across different dimensions, targeted policies and measures should promote coordinated and balanced regional development. For instance, the main city new area should boost economic vitality and digital development while strengthening coordinated and green development. Northeast Chongqing should enhance economic vitality and coordinated development. (4) Strengthen Regional Cooperation and Coordination: Regional cooperation and exchanges should be strengthened to promote the rational flow of resources, technologies, and talents. Enhancing coupling and coordination can achieve common development and mutual benefits. Government policy support should accelerate the improvement of the science and technology financial support system, fully utilizing the natural environment advantages of the "two groups" and promoting the development of advantageous industries in districts and counties. All districts and counties should deepen cooperation to share high-quality resources and complementary advantages, improving their overall economic level and narrowing regional coupling and coordination gaps. (5)Prioritize Strengthening Support for Economic Vitality, Coordinated Development, and Green Development: According to the obstacle indicators, support for economic vitality, coordinated development, and green development should be prioritized to advance the high-quality development process. Simultaneously, the importance of digital development should be recognized, with increased investment in related fields to enhance digital development. The main city metropolitan area should focus on green development support, while the "two groups" should emphasize economic vitality and coordinated development. Strengthened regional cooperation and focused development of key areas will collectively elevate the high-quality development of the regional economy.

## Supporting information

**S1 Data.**
(XLSX)

## Author Contributions

**Data curation:** Xi Zhang.

**Formal analysis:** Weijia Shu.

**Funding acquisition:** Sifang Che.

**Methodology:** Xi Zhang.

**Software:** Weijia Shu.

**Visualization:** Weijia Shu.

**Writing – original draft:** Sifang Che.

**Writing – review & editing:** Sifang Che.

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
