## [Decision Letter · Decision Letter 0]

8 Sep 2024

PONE-D-24-31019Evaluation of internal coupling and coordination degree and diagnosis of obstacle factors for high-quality regional economic development：Evidence from Chongqing's “One District, Two Groups”PLOS ONE

Dear Dr. Che,

Thank you for submitting your manuscript to PLOS ONE. After careful consideration, we feel that it has merit but does not fully meet PLOS ONE’s publication criteria as it currently stands. Therefore, we invite you to submit a revised version of the manuscript that addresses the points raised during the review process.

We look forward to receiving your revised manuscript.

Kind regards,

Jindong Chang, Ph.D.

Academic Editor

PLOS ONE

Journal Requirements:

"This work was supported by the Key Project of the National Social Science Youth Foundation project(Grant No.: 21CTJ007), the Chongqing Natural Science Foundation Project(Grant No.: cstc2021jcyj-bshX0123), and the Key Research Platform Open Project for Chongqing Technology and Business University (Grant No.: KFJJ2019030)."

Reviewers' comments:

Reviewer's Responses to Questions

**Comments to the Author**

1. Is the manuscript technically sound, and do the data support the conclusions?

Reviewer #1: Yes

Reviewer #2: Yes

2. Has the statistical analysis been performed appropriately and rigorously? 

Reviewer #1: Yes

Reviewer #2: Yes

3. Have the authors made all data underlying the findings in their manuscript fully available?

Reviewer #1: Yes

Reviewer #2: Yes

4. Is the manuscript presented in an intelligible fashion and written in standard English?

Reviewer #1: Yes

Reviewer #2: Yes

5. Review Comments to the Author

Reviewer #1: 1) Why the index system in Table 1 can be used to evaluate high-quality economic development? I don't see particularly convincing evidence for that.

2) The literature used in literature review is too old.

3) A large number of references are not searchable in the web of science, which I think is unacceptable.

Reviewer #2: Using Chongqing, an important city in western China, as an example, this article constructs a regional economic high-quality development indicator system that includes dimensions such as economic vitality, coordinated development, green development, and digital development. The entropy weight TOPSIS method, coupled coordination degree model, and obstacle factor model are applied to examine the levels of high-quality economic development, the coupled coordination degree, and the obstacles faced by Chongqing's "one district, two clusters" from 2017 to 2021. This article holds significant value for exploring the high-quality development of China's regional economy. Overall, the constructed indicators are reasonable, the methods are appropriate, and the research conclusions are highly reliable. However, there are still some issues that require further exploration, and it is recommended that revisions be made before publication.

The introduction needs further optimization. It is suggested to briefly explain why Chongqing's "one district and two clusters" was chosen as the research subject, highlighting the uniqueness and importance of this strategy for the coordinated development of Chongqing's regional economy.

The analysis of obstacle factors currently only covers the impact of primary indicators. It is recommended to refine the analysis further by examining specific deficiencies in policies, infrastructure, human resource development, technological innovation, and other relevant aspects through an in-depth analysis of specific indicators.

The conclusion could more clearly summarize the research findings and indicate the direction for future studies. It should provide an overview of the main research outcomes, the current state of regional economic development, and possible future improvement paths.

It is recommended to review the article for grammatical errors to enhance its readability.

Although this article focuses on evaluating the coupling coordination degree and diagnosing obstacle factors, there is excessive description regarding the selection of indicators, which could be condensed.

6. PLOS authors have the option to publish the peer review history of their article (what does this mean?). If published, this will include your full peer review and any attached files.

Reviewer #1: No

Reviewer #2: No

---

## [Author Response · Author response to Decision Letter 0]

29 Sep 2024

Dear reviewers:

Greetings!

Thank you very much for your meticulous review and professional comments! The problems and suggestions you have pointed out are very targeted and constructive, and the author has benefited a lot from them. Based on the review comments, the author has conducted more in-depth thinking and research on related issues, and then revised and improved the article. The following is a detailed response to your review comments, and the corresponding changes are reflected in the revised text in the revised mode.

Detailed revisions to the reviewer's comments on the article are refer to the document responses to reviewer.

---

## [Decision Letter · Decision Letter 1]

15 Oct 2024

Evaluation of internal coupling and coordination degree and diagnosis of obstacle factors for high-quality regional economic development：Evidence from Chongqing's “One District, Two Groups”

PONE-D-24-31019R1

Dear Dr. Che,

We’re pleased to inform you that your manuscript has been judged scientifically suitable for publication and will be formally accepted for publication once it meets all outstanding technical requirements.

Kind regards,

Jindong Chang, Ph.D.

Academic Editor

PLOS ONE

Additional Editor Comments (optional):

Reviewers' comments:

Reviewer's Responses to Questions

**Comments to the Author**

1. If the authors have adequately addressed your comments raised in a previous round of review and you feel that this manuscript is now acceptable for publication, you may indicate that here to bypass the “Comments to the Author” section, enter your conflict of interest statement in the “Confidential to Editor” section, and submit your "Accept" recommendation.

Reviewer #1: All comments have been addressed

Reviewer #2: All comments have been addressed

2. Is the manuscript technically sound, and do the data support the conclusions?

Reviewer #1: Yes

Reviewer #2: Yes

3. Has the statistical analysis been performed appropriately and rigorously? 

Reviewer #1: Yes

Reviewer #2: Yes

4. Have the authors made all data underlying the findings in their manuscript fully available?

Reviewer #1: Yes

Reviewer #2: Yes

5. Is the manuscript presented in an intelligible fashion and written in standard English?

Reviewer #1: Yes

Reviewer #2: Yes

6. Review Comments to the Author

Reviewer #1: The comments previously made have been greatly improved. I believe that the current level of this manuscript is acceptable for publication.

Reviewer #2: The author has revised the manuscript according to my suggestions and addressed all the concerns I raised. I recommend acceptance

7. PLOS authors have the option to publish the peer review history of their article (what does this mean?). If published, this will include your full peer review and any attached files.

Reviewer #1: No

Reviewer #2: No

---

## [Editor Report · Acceptance letter]

10 Dec 2024

PONE-D-24-31019R1 

PLOS ONE

Dear Dr. Che, 

I'm pleased to inform you that your manuscript has been deemed suitable for publication in PLOS ONE. Congratulations! Your manuscript is now being handed over to our production team.

Kind regards, 

on behalf of

Dr. Jindong Chang 

Academic Editor

PLOS ONE